# A Situational Analysis of Current Speech-Synthesis Systems for Child Voices: A Scoping Review of Qualitative and Quantitative Evidence

Camryn Terblanche [1,*], Michal Harty [1], Michelle Pascoe [1] and Benjamin V. Tucker [2]

[1] Department of Speech and Language Pathology, University of Cape Town, Cape Town 7700, South Africa; michal.harty@uct.ac.za (M.H.); michelle.pascoe@uct.ac.za (M.P.)

[2] Department of Linguistics, University of Alberta, Edmonton, AB T6G 2R3, Canada; benjamin.tucker@ualberta.ca

* Correspondence: trbcam001@myuct.ac.za

**Abstract:** (1) *Background:* Speech synthesis has customarily focused on adult speech, but with the rapid development of speech-synthesis technology, it is now possible to create child voices with a limited amount of child-speech data. This scoping review summarises the evidence base related to developing synthesised speech for children. (2) *Method:* The included studies were those that were (1) published between 2006 and 2021 and (2) included child participants or voices of children aged between 2–16 years old. (3) *Results:* 58 studies were identified. They were discussed based on the languages used, the speech-synthesis systems and/or methods used, the speech data used, the intelligibility of the speech and the ages of the voices. Based on the reviewed studies, relative to adult-speech synthesis, developing child-speech synthesis is notably more challenging. Child speech often presents with acoustic variability and articulatory errors. To account for this, researchers have most often attempted to adapt adult-speech models, using a variety of different adaptation techniques. (4) *Conclusions:* Adapting adult speech has proven successful in child-speech synthesis. It appears that the resulting quality can be improved by training a large amount of pre-selected speech data, aided by a neural-network classifier, to better match the children's speech. We encourage future research surrounding individualised synthetic speech for children with CCN, with special attention to children who make use of low-resource languages.

**Keywords:** augmentative and alternative communication (AAC); children; complex communication needs; neural networks; speech synthesis

## 1. Introduction

Children with complex communication needs (CCN) may have a developmental disorder such as autism-spectrum disorder, cerebral palsy or Down syndrome, or they may have an acquired disorder as a result of a traumatic brain injury or stroke, which results in disordered speech abilities. Children with CCN fall within a spectrum, as some present without intelligible speech whilst others have developed minimal speech, but cannot primarily rely on their speech to communicate [1]. When someone cannot rely on their own speech to communicate, they often look to other techniques to provide them with an alternative means to communicate. One such alternative is to use augmentative and alternative communication (AAC) devices. An AAC device with the capability for speech synthesis is referred to as a speech-generating device (SGD) or a voice-output communication aid (VOCA). The use of SGDs has been shown to increase the quality of life for individuals with speech impairments [2].

Without the ability to communicate, individuals may withdraw from social interaction, and even from interaction with their own family. This is often compounded when they are required to use a synthetic voice that sounds robotic, sounds like someone from a different

geographical or social background, or someone of a different age or sex [3]. In fact, it is not uncommon to see a nine-year-old girl using an adult male voice or several children in a classroom using the same voice [4,5]. Speech-synthesis technology has traditionally focused on adult speech. Developing synthetic child speech could be considered more challenging for researchers, and even more so for children with CCN who make use of low-resource languages. Despite these challenges, researchers have begun exploring new techniques and methods for natural and intelligible-sounding child-speech synthesis.

Communication is essential to learning; without access to functional communication, children with CCN are often restricted from participating in the classroom [6]. This may result in fewer instances of intentional communication, minimal language and literacy development, and poor opportunities for socialisation [1]. As a result, their potential is often underestimated. This has repercussions for other areas of their life, such as accessing healthcare, participating in family and community activities, as well as engaging in future employment [7]. Due to the importance of communication in our social, educational, professional, and personal lives, it is essential that the speech output of SGDs be as intelligible as possible. Using natural speech as the benchmark, *intelligibility* is defined as the accuracy with which an acoustic signal is conveyed by a speaker and recovered by a listener [8]. A synthesised voice with high intelligibility is thus required for the acceptability of the SGD, not only in terms of it providing the service for which is was designed, but also in terms of the positive attitudes towards the voice and the social interaction with the user [9]. In contrast to intelligibility, in which the speech signal is extracted from the context, *comprehensibility or* understandability is the degree to which speech is understood when combined with available relevant information (i.e., linguistic context or conversational topics) [8]. Speech naturalness and pleasantness could be considered important components of comprehensibility.

*Speech naturalness* can be described as how well the speech matches a listener's standards of rate, rhythm, intonation, and stress. Therefore, speech naturalness determines how natural the speech sounds to the listener and is compared to the speech they regularly hear in their immediate social environment [10]. Using a voice with a low level of naturalness, such as a robotic voice, does not match the human that it represents. Creer [9] suggested that people interact differently with machines, and if the perception is that they are addressing the communication aid rather than the human user, their interactional style will act as a further obstacle to the user's ability to form desirable social relationships with conversational partners. It appears listeners prefer a SGD to have a voice that is consistent with the characteristics of the person who is using it [2]. *Pleasantness* focuses on the pleasantness or agreeableness of the synthetic voice [10]. Although having a high quality and natural-sounding voice reduces listener fatigue and therefore contributes to positive attitudes towards the SGD by both the user and their conversational partners [9], it is also noteworthy to mention that listeners with greater exposure to synthesised speech, through practice effects, become better at analysing the acoustic–phonetic information in the signal, making better use of the acoustic cues in the synthesised speech, which results in increased intelligibility [11].

A decade ago, Yamagishi et al. [3] (p. 1) outlined that "it is not easy, and certainly not cost effective, for manufacturers to create personalized synthetic voices". However, technology has sufficiently matured to realistically emulate individual speakers' voices electronically [12]. Researchers have shown that the results of new speech-synthesis systems are good enough to mislead listeners to thinking that they are listening to authentic voices when they are in fact synthetic [13–15]. Additionally, many speech-synthesis and voice-conversion technologies have become easily accessible through open-source software, and this technology is advancing quickly. Thus, it is not improbable to consider that freely available software be used to assist individuals with CCN. However, these speech-synthesis systems are currently available for various major languages, such as English and other European languages, but are limited for South Africa's indigenous languages [10]. Accents also reflect the market size and English speech is therefore often US-accented [3].

South Africa has 11 official languages, most of which could be considered low-resource languages [16]. Thus, South Africa's rich linguistic and social diversity is not well represented in open-source technology. This is also true for commercially available SGDs, where the speech output is limited. Voices representing a *male*, *female* and in some cases, a *child* are available on the devices, but unfortunately, these same voices are used for numerous individuals and may only represent an individual's identity in a handful of cases [4].

Moreover, speech-synthesis technology has customarily focused on adult rather than child speech. Although several methods have been introduced, for a long time, researchers used speaker adaptation in conjunction with hidden-Markov-model (HMM)-based speech synthesis [17–21]. With HMM-based speech synthesis, statistical acoustic models for spectral, excitation, and duration features can be precisely adapted from an average-voice model (derived from other speakers) or a background model (derived from one speaker) using a small amount of speech data from the target speaker [22]. Novel utterances are then created when models are concatenated (overlapping and adding the signals together), generating the most suitable sequence of feature vectors from the concatenated model for which the speech waveform is synthesised [22]. Although HMM-based speech synthesis has been used for child-speech synthesis [17,19,20,23], the consensus in the community is that deep neural networks (DNNs) are more suitable for child-speech synthesis [24–32]. However, there are several difficulties associated with collecting appropriate child-speech data. Govender et al. [23] believe this is due to (a) children's short attention spans in recording sessions, (b) children's typical articulatory inaccuracies, hesitations and disfluencies, (c) children's limited reading skills as recordings are typically made from read speech, (d) children's fluctuations in emotional expression, and (e) background noise in recording environments. As children need to feel comfortable in recording settings, recording studios are often not appropriate [23]. Thus, collecting the large amounts of child-speech data necessary for child-speech synthesis is often a challenging process. Despite the challenges, the potential benefits of using this technology for children who have CCN is undeniable. This scoping review summarises the evidence base related to developing synthesised speech for children, and the results are discussed in terms of the implications for service provision for children with CCN.

## 2. Methods

### 2.1. Eligibility Criteria

There were several criteria identified for the inclusion of studies in the review. Studies included in the review were those that (a) were published between January 2006 and September 2021 and (b) included child participants or voices of children aged between 2–16 years old. Selected studies were not limited by design or language, and grey literature was included to ensure a comprehensive review. According to Colquhoun et al. [33], a scoping review outlines the research to date in a particular field of study, ultimately summarising the research findings and identifying the gaps in the literature. Although the studies may be identified systematically, a scoping review aims to identify all the relevant literature, no matter the study design and/or study quality. We chose to exclude studies older than fifteen years as speech-synthesis technology has substantially changed since then, and the voice output in SGDs often yielded poor results pre-2006, particularly considering speech naturalness and intelligibility. For example, Hoover et al. [34] reported that genuine speech was more intelligible than synthesised speech in 1987, whereas in 2020, Wang et al. [14] stated that state-of-the-art text-to-speech systems have the capability to produce synthetic speech that is perceptually indistinguishable from genuine speech by human listeners. Thus, we included current speech-synthesis systems for ecological validity and because older systems were often less intelligible, which made it challenging to generalise and compare the study results to newer systems. Finally, we chose children aged between 2–16 years to reflect the typical school-age population that would likely make use of SGDs.

## 2.2. Search Procedures

The scoping-review protocol (available on request from the corresponding author) was drafted using the preferred reporting items for scoping reviews described by Colquhoun et al. [33]. According to Colquhoun et al. [33], the steps to conducting a scoping review should follow the Arksey and O'Malley framework stages for the conduct of scoping reviews combined with the Levac et al. (2010) enhancements. The following five steps were included in the process: (1) identify the research question, clarify, and link the aim of the research with the research question; (2) identify relevant studies by balancing feasibility, breadth and comprehensiveness; (3) select studies using an iterative team approach to study selection and data extraction; (4) chart the data using a descriptive analytical method; (5) collate, summarise and report the results [33].

A composite search strategy was executed to ensure all studies meeting the selection criteria were identified, whilst avoiding a biased evidence base. The search strategies were drafted by the first author in consultation with a colleague with expertise in electronic-search strategies and further refined through team discussion. The databases and search terms used are outlined in Table 1. Firstly, electronic searches of several databases were conducted using the search terms presented in Table 1. Next, ancestral searches of references cited in studies that met the selection criteria were conducted, which subsequently yielded additional journals for electronic searching. Journal articles, book sections, conference proceedings, conference papers, thesis papers and reports were included in the search. Irrelevant publications, duplicated publications and publications that did not meet the inclusion criteria were removed. Following this, the final search results were exported into EndNote.

**Table 1.** The search procedures used in the scoping review.

| | |
|---|---|
| **Databases** | • EBSCO Host<br>• Scopus<br>• PubMed<br>• Google Scholar |
| **Sources of evidence** | • Electronic databases<br>• Reference lists<br>• Electronic searching of key journals<br>• Existing networks, relevant organisations, and conferences |
| **Search terms** | (child* AND ("speech synthesis" OR "synthetic voice*" OR "speech synthesi?er" OR "digiti?ed speech")) |
| | (child* AND VOCA AND ("digiti?ed speech" OR "synthesi?ed speech" OR "speech synthesis")) |
| | (child* AND "speech generating device" AND ("digiti?ed speech" OR synthesi?ed speech" OR "speech synthesis")) |

Two reviewers independently screened the 217 publications using Rayyan [35] to reduce bias and improve reliability. Both a PhD student in speech and language pathology and a reliability agent, a professional with a doctorate in speech and language pathology, electronically searched the journals and selected relevant studies. Reviewers evaluated the titles, abstracts, and then the full text of all publications identified by our searches for potentially relevant publications. Researchers resolved disagreements on study selection and data extraction by consensus and discussion with other reviewers if needed. Inter-rater reliability was 96.6% for the search procedures (agreements divided by agreements plus disagreements with the outcome multiplied by 100).

## 2.3. Coding Procedures

A data-charting form was jointly developed by two reviewers to determine which variables to extract. The data-charting form was piloted with several abstracts and then amended to more accurately capture elements of the speech-synthesis systems. The revised

data-charting form was used to chart full-text data. Data charting was independently performed by one reviewer and 15% of the extracted data were audited by a second reviewer, who has their doctorate in speech and language pathology. The results were discussed between reviewers, and discrepancies resolved via consensus. Each study was coded with respect to (a) the aim of the study, (b) the design, (c) the voice-output language, (d) the study population and sample size (i.e., sample size, sex, age, typically developing children/children with disability), (e) the method (novel synthesis system (i.e., hidden-Markov-model-based synthesis, direct waveform concatenation, etc.), commercial SGD (i.e., commercial software and application voices) and review (i.e., scoping review)), and (f) the outcomes.

### 3. Results

In total, 58 studies were identified for the review (See Supplementary Materials). Figure 1 depicts the scoping-review process. Of the 58 included studies, 2 articles were reviews, 5 articles discussed commercially available synthesis systems, and the remaining 51 articles discussed novel synthesis systems. A total of 31 studies focused on English voices, 21 studies discussed languages other than English, and 9 studies did not specify the language used (some studies considered more than one language).

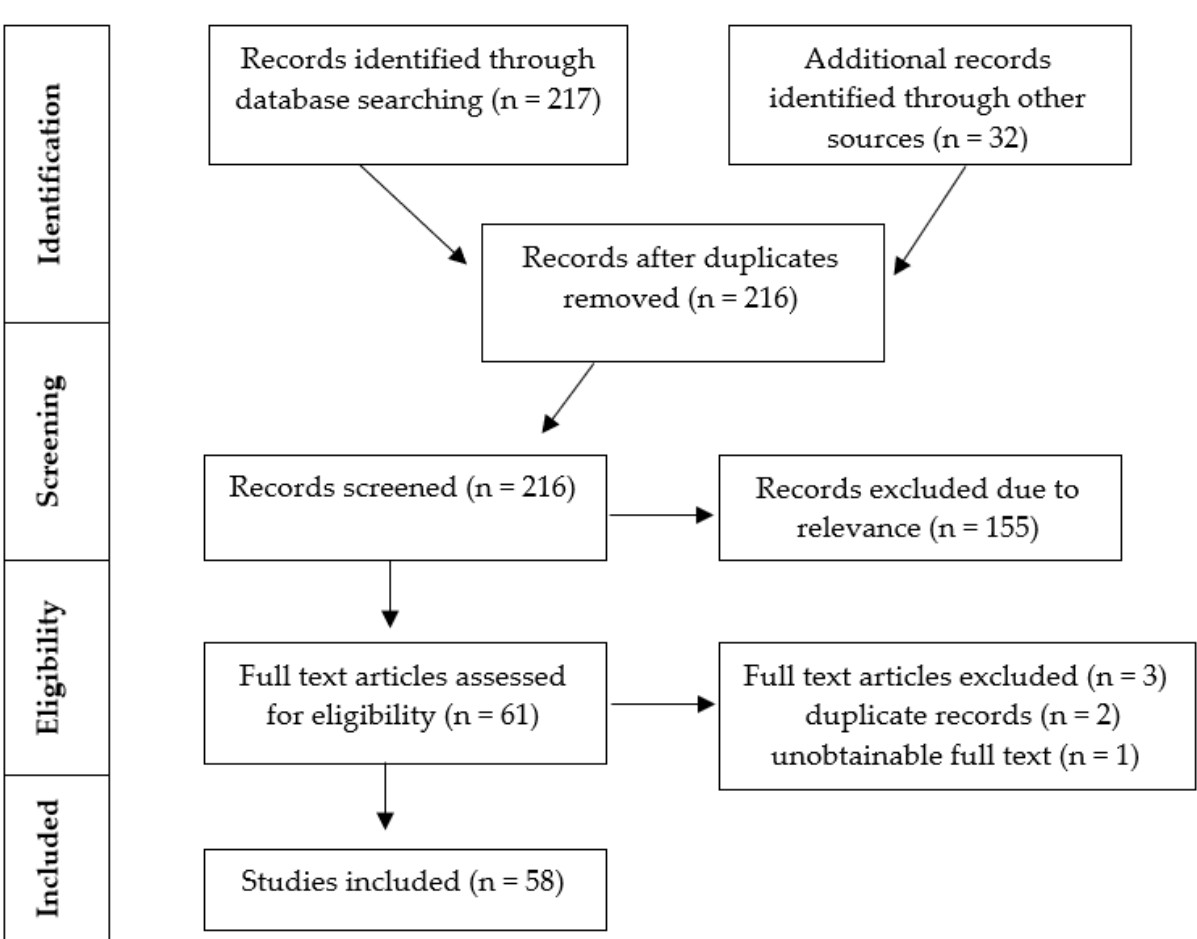

**Figure 1.** A PRISMA flow diagram depicting the scoping-review process.

### 3.1. Language

Even though the review did not exclude studies by language, it was unsurprising that English was most often used in commercially available voices, such as DECtalk and VeriVox [36]. It was also the language most often selected to create synthesised speech for children. Accents often reflect the market size [3], which is likely why the synthesised

speech was most often US-accented [17,18,23,25,37–40]. However, it was refreshing to discover studies focusing on other variants of English such as Irish English [41] and Indian English [19,42]. Although less common, researchers also considered other languages when experimenting with child-speech synthesis, including Norwegian [5], Spanish [18], Punjabi [43], Finnish [44], German [45,46], Czech and Slovak [47], Mandarin [27,31] and quite often, Italian [21,29,30,48–52].

### 3.2. Speech-Synthesis Systems

Prior work has focused on speaker adaptation in conjunction with hidden-Markov-model (HMM)-based speech synthesis [17–21]. In one study, voices created by the HMM-based systems were generally regarded as more similar to the original speaker than the voice-converted unit-selection systems [20]. Researchers concluded that although the child-speech data had poor coverage of the phonetic/prosodic units of the language, an inconsistent reading style and imperfect recording conditions, it was feasible to build child voices by using the HMM-based speech-synthesis framework [20]. Similarly, the HMM-prototype voice created in another study had challenges related to intonation and pronunciation, noise levels, naturalness and volume levels [5], but the voice created was regarded to be around seven years old, which suggests the method can build child voices. Jia, Zheng, and Sun [53] recently attempted to synthesise emotional children's speech by comparing three models (1. HMM model, 2. conditional GAN model, 3. current model CycleGAN). It was clear that the emotional-classification performance of the CycleGAN model was the best, and the accuracy of the HMM-model was the lowest. It appears that although the HMM may be a feasible method for child-speech synthesis, findings suggest that the naturalness of the synthetic speech may be compromised.

Modern synthesis methods, specifically related to deep learning, have had more success at improving the quality of child-speech-synthesis models. Traditionally, researchers have focused on Gaussian mixture models–hidden Markov models (GMM–HMM) for synthetic child-speech development, although more recently, there has been consensus in the community that deep neural networks (DNNs) are suitable for child-speech synthesis [24–32], and results show that DNNs have outperformed the older Gaussian mixture models (GMMs) [37,54]. According to Shivakumar and Georgiou [37], the success of DNNs is due to their ability to use large amounts of training data. Additionally, human speech is filled with non-linearity, and DNNs are able to better approximate the non-linear functions that are needed to model speech [37]. Metallinou and Cheng have gone so far as to say that DNN acoustic models work better than GMMs, even when GMMs are trained with approximately eight times more data [54]. Both Cosi [30] and Giuliani and BabaAli [29] used a hybrid DNN–HMM-based automatic-speech-recognition system. They had approximately 10 h of Italian child speech, and results showed improvements over the traditional GMM-based systems.

In addition, using a combination of adult and child training data in the DNN-based models resulted in improved results [24–26]. In particular, combining child speech with adult female speech has been proven to be advantageous [25,55]. As the length of an adult female vocal tract is closer to the length of a child's vocal tract in comparison to that of males, manipulating an adult female voice into a child's voice has often been more successful [25]. In fact, the child voice that was adapted from an average-male-voice model experienced significantly larger distortion than that adapted from the average-female-voice model [55]. Similarly, a model trained on a male voice resulted in a less naturalistic child voice [47,56]. Additionally, in order to improve the acoustic mismatch that often occurs as a result of using adult training data on child models, pitch normalisation has been introduced. It has been shown that significant improvement can be obtained with the maximum-likelihood-based explicit pitch normalisation of children's speech [57–62].

There are several other adaptation techniques that could be utilised to account for the mismatch, and these techniques include, although not exhaustively: adapting acoustic models with maximum-likelihood linear regression (MLLR), constrained structural maximum a

posteriori linear regression (CSMAPLR), maximum a posteriori (MAP), speaker-adaptive training (SAT) based on constrained MLLR (CMLLR), and vocal-tract-length normalisation (VTLN) [37]. VTLN aims at reducing the interspeaker and inter-age-group acoustic variability due to vocal-tract-length (and shape) variations among speakers by warping the frequency axis of the speech-power spectrum [49]. There are two ways to account for the differences in vocal-tract length between adults and children. One could apply VTLN to adult utterances during training to make the normalised features more like children's speech. Alternatively, one could apply VTLN to the child's utterances during training and testing to make them more similar to the adults' speech [25]. Saheer et al.'s [55] study shows that combining VTLN with CSMAPLR resulted in an improved adaptation method for both HMM-based automatic speech recognition and text-to-speech. It was clear that when there was a limited amount of adaptation data available, even as little as one sentence, the VTLN yielded the best naturalness and intelligibility results. When only one sentence was used, the CSMAPLR transformation was not intelligible at all. This is an indicator that VTLN is useful in child-speech synthesis. Although VTLN has been recommended for children's speech, it has also been suggested that it should be conducted differently for children when compared to adult speech [63]. The results from Umesh, Sinha and Rama Sanand's study [63] suggested that unlike conventional VTLN, it was better not to scale the bandwidths of the filters, but rather to scale the filter centre frequencies [63].

Interestingly, aside from creating a child voice by adapting an average adult voice or an average child voice, Karhila et al.'s study [44] compared two additional adaptation methods using stacked transformations: StA and StVA. In the first method, StA, an average voice trained from adult data was adapted using training data of the average child voice. The resulting synthetic voice sounded child-like with regards to pitch, pronunciation, and rhythm. This model was then further adapted to resemble a specific target child speaker. In the second method, StVA, an average voice trained from adult data was adapted using training data of the average child voice, and then VTLN occurred [44]. It was found that stacked transformation systems (StA and StVA) were preferred by listeners and resulted in better adapted voices than directly adapting the average adult voice or the child voice [44].

### 3.3. Child-Speech Data

Due to the scarcity of child-speech data available for researchers, along with the typical articulatory errors, simplifications, hesitations, and disfluencies present in child speech, researchers have attempted to overcome these difficulties when developing child-speech synthesis, in a number of ways. Serizel and Giuliani [28,32] used a similar technique to Tong et al. [27] to account for limited training data. Hasija, Kadyan, and Guleria [43] decided to account for the scarcity of available child speech by combining two corpora, one synthesised and one clean (authentic), in order to generate high-quality synthetic child speech. Thereafter, a corpus with a greater quantity was used. Their results indicated that the merged-data corpus showed a reduced word-error rate of the automatic-speech-recognition system with a relative improvement of 9–12%. In a study by Govender, Nouhou, and De Wet [23], data were selected from an automatic-speech-recognition corpus to build child voices instead of using text-to-speech data. Speech recordings used for automatic-speech-recognition corpora are usually shorter and more spontaneous in comparison to the carefully articulated recorded speech used for text-to-speech development. The criteria considered for adaptation in their study [23] included: (1) clean data (with regards to transcription) and no mispronunciations or mistakes in the recordings, (2) data including mispronounced words, (3) number of words in the utterance, (4) rate of speech, and (5) maximum fundamental frequency [23]. The results showed that when comparing intelligibility, the word-error rates were not as close to voices that were built using speech data that were specifically recorded for speech-synthesis purposes, such as text-to-speech data. However, in terms of naturalness, if data were selected according to particular criteria, then automatic-speech-recognition data could be used to develop child voices that are comparable to text-to-speech voices [23].

In addition, many researchers have attempted to build child-speech models by adapting adult-speech models. This has been proven to be a viable method when there are limited speech data available [5,18,23–26,37,44,48]. In a study by Hagen, Pellom and Hacioglu [18], a synthetic children's model was derived without child-speech data by using adult-speech data. While they assumed that the availability of child-speech data would have improved the resulting acoustic models, the approach was effective when child-speech data was not available [18]. According to a number of researchers [19,37,43], increasing the amount of training or adaptation data results in lower word-error rates. Thus, when creating child voices from adult-speech models, it appears that using more data yields better results, but data should not simply be selected blindly. Some researchers have recommended that one should select training speakers that are closer to the target speaker to train the initial models [23].

Shivakumar and Georgiou [37] have concluded that any amount of child data is helpful for adaptation. In their experiments, even as little as 35 min of child-adaptation data were found to yield relative improvements of up to 9.1% over the adult model [37]. However, researchers still need to carefully select the child-speech data, as the typical articulatory errors and disfluencies that are commonly present in child speech will also appear in the synthetic speech without careful selection. This was the case in Kumar and Surendra's study [19], and although the output sounded like child-read speech, it was not fluently spoken, which is not ideal for a SGD. Of course, if there are enough child-speech data available, a child voice can be created without directly adapting an average adult voice, as was the case with Karhila et al.'s study [44]. However, it was found that the adaptation of an average adult voice was preferred to the adaptation of the child voice when there were enough adaptation data. In contrast, when there were very little data available, the child voice was preferred over the average adult voice [44]. This appears to support the premise that more data are not always preferred when proper selection does not occur.

When an average voice is created (which is derived from many speakers), one needs to determine how the voices are going to be clustered. In Govender, Nouhou, and De Wet [23], it was shown that using a gender-independent average-voice model (male and female) resulted in higher quality synthetic child speech than using a gender-dependent average-voice model (male or female). In Watts et al. [20], two gender-dependent average-voice models were first trained using speaker adaptive training (SAT). Following this, the parameters of both gender-dependent models were clustered and tied using decision-tree-based clustering, where gender was included as the context feature. Lastly, the clustered HMMs were re-estimated using speaker adaptive training, with regression classes for the normalisation being determined from the gender-mixed decision trees [20]. However, in contrast, gender-dependent average-voice models were used by [25,55] where improved performance, as compared to the baseline, was reported.

### 3.4. Intelligibility

It is clear from the data gathered that the speech output of commercially available devices is limited. The speech output of commercially available devices does not often reflect the user's language, age, sex, or personality [4,36,64,65]. In a study by Begnum et al. [5], parents of children using SGDs responded positively towards voices that they believed matched their child's identity, particularly as they were sex and age appropriate. However, as the synthesised prototype child voice had flawed intonation and periods of unintelligibility, they reported that they would rather opt for an adult voice that was clear and easy to understand. This was supported by the teachers who needed to understand the voice in demanding surroundings [5]. Comparatively, one of the child users stated that he would have liked to use the child voice on his SGD, but that he did not like the sound of the prototype voice created [5]. Begnum et al. [5] found that both the children and their communication partners would prefer a child voice that matches the child's vocal identity but not at the cost of intelligibility. A higher-quality speech output may be more important than matching the child's vocal identity [5].

There are additional factors that may affect the intelligibility of the speech output. According to a study by Drager et al. [66], two contextual variables (words vs. sentences and topic cues vs. no topic cues) interacted with speech type (digitised vs. synthesised speech). Listeners experienced increased intelligibility for contextual words and sentences and in particular, increased intelligibility of sentences compared to single words. In other words, children who communicate with other children that make use of a single word AAC device may find it difficult to understand as they have little contextual information to point them in the right direction [66]. This was also found in other studies [36,67] as listeners reported increased intelligibility of words when words where embedded in sentence utterances rather than in isolation.

### 3.5. Age

Although technology has changed substantially since 2009, results from Von Berg et al.'s [36] study suggested that child voices in the commercially available systems, DECtalk and VeriVox, were significantly less intelligible than the adult voices in the same commercially available systems, for both single-word and phrase-level intelligibility tasks. Similarly, Shivakumar and Georgiou [37], conducted an analysis of large-vocabulary continuous-speech-recognition (LVCSR) adaptation and transfer learning for children's speech, using five different speech corpora. The trend showed that as the age of the child increased, a smaller amount of adaptation data was required. The overall performance increased as the child's age increased, irrespective of the adaptation configuration. Younger children therefore needed more data to reach the same level of performance as older children. In other words, older children had a decreased word-error rate as opposed to younger children [37]. In another study focusing on child-speech synthesis, various acoustic adaptation and normalisation techniques were implemented. The word-error rate decreased with age from 6 years to 11 years, resulting in an approximate linear increase in performance with age classes [68]. Despite this, Drager and Finke [69] found that a child speaker's original speech intelligibility and articulation skills may be better indicators than age. Some discretion should be applied, as on occasion, a four-year-old child may present with fewer articulatory errors than a seven-year-old child, and therefore appear more intelligible, despite the difference in age.

## 4. Discussion

This scoping review addressed the current state of knowledge regarding the development of child-speech synthesis. Based on the reviewed studies, it is clear that child-speech synthesis is still a growing field. However, relative to adult-speech synthesis, developing child speech is notably more challenging for researchers. It is even more challenging when one considers creating synthetic child voices for children with CCN, particularly for those speaking low-resource languages. Thus, these findings are considered in terms of the implications for service provision for this group of individuals.

### 4.1. Language

The review shows that English, particularly US English, is frequently used as the language of choice in child-speech-synthesis systems. However, English is often used due to the market size [3] and the availability of English data required for training. Good-quality recordings, in addition to the phonetic and linguistic knowledge that is required for the annotated text resources in the language, can come at a high cost and unfortunately, they do not exist for some languages [70]. Some researchers have attempted to overcome these difficulties in adult-speech synthesis by crowdsourcing speech samples [71], using available data such as audiobook data [16] and by using nonideal corpora. In some cases, where a bootstrapping technique can be used, data from a major language can be shared with data from a new language if the new language is comparable with a major language. In those cases, researchers can create synthetic voices for low-resource languages with a small amount of training data [72]. For example, since Mandarin and Tibetan belong to

the Sino-Tibetan language family, the speech data and speech-synthesis framework can be borrowed from Mandarin, whilst only making use of a small amount of Tibetan training data [72]. This could potentially be a useful technique for the building of synthetic voices in the Bantu languages of South Africa. In a recent study, results were encouraging when working with data from children speaking in a second language or a non-native language, i.e., Italian children speaking both English and German and German students speaking English [73]. A multi-lingual data adaptation in transfer learning and multi-task learning framework was found to be useful [73]. Nevertheless, there are an overwhelming number of languages that have yet to be considered for speech synthesis.

As English is often prioritised for speech synthesis, children who do not speak English, along with those who do not use it as a first language, are often disadvantaged. This is particularly apparent in South Africa because if a child has to use a pre-loaded voice through an AAC device, English could very well be the child's second or third language [7]. As children with CCN may also have comorbid language difficulties, the limited language options subsequently place an additional barrier on their communication. It is well established that an individual's voice is unique and can signify particular elements, such as their physical size, age, sex, race, intellectual ability, geographical and social background, as well as their personality [4,64,74]. In many countries, including South Africa, a person's language also plays a role in their identity by representing their culture. Thus, the effect that a language selected for an SGD could have on a child's identity should not be minimised. It is also important to remember that for children with CCN who need to make use of SGDs, speech synthesis not only forms the basis of their personal identity, it is also crucial for communication and an essential component of social interaction [3]. If the language on their device is different from those in their immediate social environment, then the child's communication effectiveness is negatively influenced.

*4.2. Speech-Synthesis Systems*

When considering building child-speech-synthesis systems, one needs to ask several questions: (a) What kind of average-voice model (gender-dependent or gender-independent) is the most appropriate initial model from which adaptation will occur? (b) Are there enough training data for the initial model? (c) Are there enough adaptation data to improve adaptation performance? (d) Should some training/adaptation data be excluded from the model? If so, how much? (e) What kind of adaptation techniques (VTLN, SAT etc.) will be utilised? If so, would it be conducted on the adult speech, the child speech or both? (f) What kinds of transform functions are appropriate?

It appears that there are numerous synthesis methods available, each with their own advantages and disadvantages. As mentioned previously, neural-network-based text-to-speech systems have been gaining popularity in recent years. This speech-generation method is based on deep learning and it can mine the potential correspondence from multiple corpora and automatically learn the dependence from the source sequence to the target sequence [53]. A neural-network-based acoustic model predicts a sequence of acoustic features, including the mel-cepstral coefficients (MCCs), interpolated fundamental frequency and voicing flags. Once completed, a vocoder analyses and converts these MCCs and the fundamental frequency into a waveform, which forms a synthesised voice [14,43]. According to Wang et al. [14] and Terblanche, Harrison, and Gully [15], new neural-network-based text-to-speech systems, such as Tacotron 2, produce synthetic speech that has a perceptually high level of naturalness and good similarity to adult target speakers. Tacotron was also used by Hasija, Kadyan, and Guleria [43] for the development of children's synthetic speech. DNNs are therefore suitable for child-speech synthesis [24–32] and often outperform the older GMM models [37,54]. However, it should be noted that a large amount of training data is not always freely available when developing child speech, particularly when one considers building child speech for low-resource languages. It appears that in order to account for this, researchers began using both adult and child speech for training in the DNN-based models, and this resulted in improved results.

There are a number of techniques that have been introduced to improve child-speech synthesis. Firstly, researchers need to define better acoustic features for children's speech. The most commonly used features include mel-frequency cepstral coefficients (MFCCs), filterbank, and perceptual-linear-prediction (PLP) coefficients [25]. Typically, MFCCs achieve the best performance in GMM-based systems [59,68] while mel-filterbank coefficients are often used in DNN-based systems [25]. Secondly, due to the differences in child-speech development, pronunciation modelling is required [68]. Thirdly, VTLN is often used to account for the differences in vocal-tract length between speakers [44,50,55,75]. Fourthly, modal-adaptation techniques are often used, such as maximum a posterior (MAP) and maximum-likelihood linear regression (MLLR) [25,49].

### 4.3. Child-Speech Data

As previously discussed, collecting child speech presents with many difficulties. The type of child speech typically available in corpora does not always provide complete coverage of all the speech units in the language, and it is often inconsistently read and imperfectly recorded [19,23]. According to a number of researchers [19,37,43], increasing the amount of training or adaptation data results in lower word-error rates, indicating greater intelligibility, even if imperfect data are included [23]. This is in contrast to another study, where researchers concluded that using fewer data of superior quality is preferred in adult-speech-synthesis models, as opposed to using more data of inferior quality [16]. Although fewer data are available, the high-quality data reportedly result in adult voices of improved naturalness and intelligibility [16].

Comparably, when adapting adult-voice models to resemble child speakers, some researchers suggest clustering the data (i.e., age, sex, max fundamental frequency, etc.) so as to develop an average-voice model that closely matches the target child speaker to some degree [17,64], while other researchers suggest adaptation can be performed after the average-voice model has been trained, with adaptation techniques such as VTLN [44,50,55]. In a study by Yamagishi et al. [22], it was seen that the greater the difference between the average-voice model and the target speaker, the poorer the resulting target voice quality would be. This finding suggests that simply using an average adult voice for adaptation to a child target speaker will yield poor results. However, it is quite clear from the review that this method has been used on numerous occasions [5,18,23–26,37,44]. In order to pre-empt the decreased quality due to the mismatch between adult and child, researchers have suggested selecting training speakers that are closest to the child target speaker. The resulting quality could be improved by training a large amount of pre-selected data, aided by a neural-network classifier, to better match the children's speech [40]. Training speakers to resemble the child's vocal quality to some degree could either be done in the initial training phase by selecting similar training speakers, or after data normalisation has occurred. In addition, if the average adult voice is further augmented with a small amount of children's speech, a closer match may be found [37,43].

### 4.4. Intelligibility

Because of the lack of redundant auditory and visual cues found in synthesised speech, listeners must allocate more attentional resources to process synthesised speech, as compared to natural speech [76]. These factors, such as a lack of visual cues, are likely to influence the comprehensibility of any communication mode (natural or synthesised), when the speaker cannot be seen, such as what one might experience with telephone speech [77]. Similarly, natural speech is often comprehended faster than synthesised speech, especially when there is a high level of background noise or the listener's attention is divided [76]. However, as the quality of synthetic speech improves, the margin between synthetic speech and natural speech decreases, as does the cognitive load required [9]. In terms of what individuals prefer, it appears that people prefer listening to voices that match the vocal identity of the user [5,9,78]. Despite this, there is evidence to suggest that individuals

would rather choose high intelligibility of the speech output instead [5,77,78]. A high level of intelligibility is therefore crucial in the development of synthetic speech.

It has been shown that human listeners adjust in subtle but systematic ways to understand synthetic speech. Understanding synthetic speech often requires a greater cognitive load than understanding natural speech, but this cognitive load decreases when the listener becomes accustomed with the voice [9,76]. For children and adults, comprehensibility of the synthesised speech signal often improves after greater exposure [45]. This is also true for individuals with intellectual disabilities, as studies showed that their perception of the synthetic speech improved after systematic exposure to it [11,65].

Results have suggested that context and the length of the utterance play a role in the intelligibility of synthesised speech [66,76]. In other words, longer utterances are more intelligible than single words, unless the listeners are given a closed response set (i.e., having a set of predefined answers and pointing to a picture in response to a stimulus) [11,65]. Longer utterances typically contain more linguistic context than single words, which listeners can use to increase signal-independent information. This is clinically important as some children with CCN may have comorbid difficulties, such as decreased working memory and impaired language skills, resulting in decreased use and comprehension of longer utterances [1]. Thus, single-word AAC devices are often common starting points for children with CCN. However, if their speech output is supported with a symbol or picture on their AAC device, it may assist their communication partners in understanding them. It is well-recognised that successful AAC communication depends on both the AAC user and their communication partners [2,8,79,80].

Intelligibility of the speech output on a speech-generating AAC device is essential for several reasons. If children are unable to communicate with their natural voice, the speech output will assist them in learning how to use the AAC system; it provides feedback to the child and their communication partner and it allows for successful interaction opportunities with new communication partners [66]. Moreover, the speech output may provide additional verbal modelling, which could result in an increase in the child's spontaneous speech production [81]. Researchers have discussed how sound produced by an iPad or a SGD may act as a reinforcer for a child user, which subsequently motivates them to use it to communicate [81–83]. In one study, children with intellectual disabilities were also able to generalise their knowledge of the acoustic-phonetic properties of synthetic speech to novel stimuli [65]. This clearly has some vitally important clinical implications for children with intellectual disabilities and would likely result in an increase in their participation in academic environments.

### 4.5. Age

Typically, adults are considered more intelligible than children due to the expected patterns of speech simplifications and articulatory errors often observed in a developing child's speech. Along with the physical, linguistic, and prosodic differences observed in children's speech [37,43], it is not surprising that the acoustic and intelligibility mismatch between adult and child speech is also considered a challenge in the development of synthetic speech for children. It is well established that characteristics of speech, such as pitch, formant frequencies and phone duration are related to the age of the speaker [50]. These acoustic differences result from children having shorter vocal tracts and smaller vocal folds than adults [84]. Younger children show higher pitch and fundamental frequency in comparison to older children [85]. In Shivakumar and Georgiou's study [37], the trend showed that as the age of the child speaker increased, less data adaptation was required. Older children did not experience as great of a mismatch between the adult speech (which was used as training data), while younger children showed that considerably more data were needed to account for the mismatch [37]. Younger children therefore show a considerably higher amount of intra- and inter-speaker variability as compared to older children and adults [49,51]. Younger children have high levels of acoustic complexity, which can be attributed to "three main factors (i) shifted overall spectral content and formant frequencies

for children, (ii) high within-subject variability in the spectral content that affects formant locations and (iii) high inter-speaker variability observed across age groups, due to developmental changes, especially vocal tract" [37] (p. 2). Thus, more parameters were required to accurately capture the increased complexity in their speech [37]. If one were to consider building a young child voice, with its high level of acoustic complexity, it may be prudent to first consider the availability of the data necessary for adaptation. Although there is not a consistent cut-off age for the most suitable levels of intelligibility [76] (i.e., seven-year-old speech vs. four-year-old speech), it appears that as a child's level of intelligibility increases and their acoustic variability decreases, which may be at approximately nine years old, the less challenging it would be to build a synthetic voice for them.

## 5. Conclusions

This scoping review addressed the current state of knowledge regarding the development of child-speech synthesis based on research conducted over the last 15 years and reveals potential directions for future research. However, the possibility of not including all relevant articles must be recognised. Selecting other databases may have identified additional relevant studies. Additionally, relevant articles may have used terms other than speech-generating device or voice output communication aid. Finally, we did not critically appraise individual sources of evidence, as the focus of this review was to identify available evidence rather than to evaluate it.

This scoping review did not evaluate specific speech-synthesis systems or judge the utility of these systems for children with CCN. However, what emerges from the evidence is that speech-synthesis technology has improved remarkably over the last 15 years. In fact, in the last few years, it has become possible to create intelligible and natural-sounding synthetic speech that has the potential to mislead listeners to thinking that they are listening to authentic speech. Many speech-synthesis and voice-conversion technologies have become easily accessible through open-source software. However, based on the studies reviewed, relative to adult-speech synthesis, developing child-speech synthesis, particularly for young children, is notably more challenging for researchers. Child speech often presents with acoustic variability, disfluencies, and articulatory errors. In addition to this, it is often challenging to collect child speech due to children's short attention spans, limited reading skills and the diverse recording environments.

To account for these challenges, numerous researchers have attempted to adapt adult-speech models, using a variety of different adaptation techniques. In most cases, adult-speech data are used in combination with a small amount of child-speech data to create a synthetic child-like voice. Adapting adult speech has proven successful in child-speech synthesis and it appears that the resulting quality can be improved by training a large amount of pre-selected speech data, aided by a neural-network classifier, to better match the children's speech.

For children who are unable to communicate using their natural speech, speech synthesis could provide a viable means of communication. The selected synthetic voice used in the speech-generating device is therefore likely to become an extension of themselves. With that in mind, we propose that the synthetic voice be individualised to best represent the child's vocal identity, with regards to at least their language, sex, and age. With an individualised synthetic voice, children with complex communication needs could potentially increase their intentional communication skills, participation, language, and literacy skills in a classroom setting. As multiple children in one classroom may need a synthetic voice, having an individualised voice would benefit them greatly as teachers would be able to differentiate speakers in class, there may be greater technology-adoption rates and an increased level of socialisation between children. Future research could focus on developing a system that is acceptable to the child and improves its performance over time based on continued use by the child.

As language is a large part of an individual's vocal identity, the language selected for the device is another important element to consider. Many of the speech-synthesis systems

are usually designed for major languages, such as English, but are limited for low-resource languages. Promisingly, it was discovered that if there are enough training data available, either collected in the typical fashion or atypically, through crowdsourcing, or by combining language-similar corpora (borrowing speech data from other corpora that fall within the same language family), one should be able to create natural and intelligible synthetic speech for children in any language. We therefore believe that future research should investigate individualised synthetic speech for children with complex communication needs, with special attention to children who make use of low-resource languages. It would be interesting to determine if the residual speech produced by a target child with complex communication needs could be combined with speech data from typically developing children and utilised to develop a unique and individualised synthetic voice for a particular target child, whilst making use of open-source speech-synthesis software.

**Supplementary Materials:** The following supporting information can be downloaded at: https://www.mdpi.com/article/10.3390/app12115623/s1.

**Author Contributions:** Writing—original draft preparation, C.T.; writing—review and editing, M.H.; writing—review and editing, M.P.; writing—review and editing, B.V.T. All authors have read and agreed to the published version of the manuscript.

**Funding:** The financial assistance of the National Research Foundation (NRF) of South Africa (DSI-NRF Reference Number: MND200619533947) is hereby acknowledged.

**Institutional Review Board Statement:** Not applicable.

**Informed Consent Statement:** Not applicable.

**Data Availability Statement:** Not applicable.

**Conflicts of Interest:** The authors declare no conflict of interest.

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
