# Peer review of "A Situational Analysis of Current Speech-Synthesis Systems for Child Voices: A Scoping Review of Qualitative and Quantitative Evidence"

_applsci, doi:10.3390/app12115623_

Round 1

Reviewer 1 Report

Thank you for the opportunity to review: A situational analysis of current speech synthesis systems for child voices: A scoping review of qualitative and quantitative evidence

I found the manuscript overall well written and very interesting and will provide an important contribution to the field.

I have two minor suggestions for the introduction and some requests to clarify the scoping review methodology in the methods and result section where I have used the PRISMA Extension for Scoping Reviews (PRISMA-ScR): Checklist and Explanation as a reference for my suggestions. https://www.acpjournals.org/doi/10.7326/M18-0850

Line 53-55 I suggest moving “This scoping review summarises the evidence base related to developing synthesised speech for children and the results are discussed in terms of the implications for service provision for children with CCN” to the end of the introduction and then omit line 121-123 “The remainder of this paper is laid out as  follows: Section 2 describes the method used in the scoping review, Section 3 presents the results which are then discussed in Section 4, and conclusions are presented in Section 5” which is a given for most publications.

1.      Does a protocol or registration of the scoping review exist? If not, just a quick statement would be helpful. If it does, please include the registration number.

2.      Please expand on the process for selecting sources of evidence (i.e., screening and eligibility) included in the scoping review

3.      Please expand the methods of charting data from the included sources of evidence (e.g., calibrated forms or forms that have been tested by the team before their use. The inter-rater reliability to select publications have been described, however please clarify whether data charting was done independently or in duplicate) and any processes for obtaining and confirming data from investigators, in addition to the information you already provided.

4.      In the Result section, please expand to include numbers of sources of evidence screened, assessed for eligibility, in addition to the number you mentioned included in the review, with reasons for exclusions at each stage, ideally using a flow diagram.

5.      Please include a small paragraph on potential limitations you identified throughout with the Scoping review process

Reviewer 2 Report

This is a diligent and thorough review of work on synthesising children's speech. It will be useful for someone who is entering this field but it won't help such a person to understand how speech synthesis works. Pesumably that is what is meant by a 'scoping review'.

Complicating factors are that

The current state of the art is confusing, with different techniques applied to different tasks within the genertal field. In contrast to many topics within speech technology there does not seem to a a 'challenge', i.e. a common ask which different tlabs submit entries too.

The authors do not seem to be experts in the technology and hence findit difficult to pick out important differences in systems and construct arguments. For instance 'Neural Nets' are referred to many times but the different types of network, and their pros and cons, are never articulated. Similarly, the possible combination of statistical nd connectionist architectures is mentioned by never explained.

The way the technology is deployed is as important as the technology itself. Just measuring speech intelligibity doesn't tell you much. A system which improves performance as an individual child uses it, which learns to belong to the child and is accepted by the child, would haver meny advantages.
